# Escape Sky-high Cost: Early-stopping Self-Consistency for Multi-step Reasoning

Yiwei Li[1][†], Peiwen Yuan[1][†], Shaoxiong Feng[2], Boyuan Pan[2], Xinglin Wang[1],
Bin Sun[1], Heda Wang[2], Kan Li[1][‡]

[1] School of Computer Science, Beijing Institute of Technology
[2] Xiaohongshu Inc

{liyiwei,peiwenyuan,binsun,wangxinglin,likan}@bit.edu.cn
{shaoxiongfeng2023,whd.thu}@gmail.com  {panboyuan}@xiaohongshu.com

## Abstract

Self-consistency (SC) has been a widely used decoding strategy for chain-of-thought reasoning. Despite bringing significant performance improvements across a variety of multi-step reasoning tasks, it is a high-cost method that requires multiple sampling with the preset size. In this paper, we propose a simple and scalable sampling process, **E**arly-**S**topping **S**elf-**C**onsistency (ESC), to greatly reduce the cost of SC without sacrificing performance. On this basis, one control scheme for ESC is further derivated to dynamically choose the performance-cost balance for different tasks and models. To demonstrate ESC's effectiveness, we conduct extensive experiments on three popular categories of reasoning tasks: arithmetic, commonsense and symbolic reasoning over language models with varying scales. The empirical results show that ESC reduces the average number of sampling by a significant margin on six benchmarks, including MATH (-33.8%), GSM8K (-80.1%), StrategyQA (-76.8%), CommonsenseQA (-78.5%), Coin Flip (-84.2%) and Last Letters (-67.4%), while attaining comparable performances [*].

## 1 Introduction

Large language models (LLMs) have exhibited strong reasoning capabilities (Bubeck et al., 2023), especially with chain-of-thought (CoT) prompting (Wei et al., 2022). Based on this, Wang et al. (2023) introduce a simple decoding strategy called self-consistency (SC) to further improve reasoning performances, leveraging the fact that complex reasoning tasks typically allow for more than one reasoning paths leading to the correct answer. In contrast to the standard chain-of-thought prompting which only generates the greedy one, this method samples multiple reasoning paths according to the predetermined sample size, and then derives the final answer through voting-based scheme.

However, despite generally leading to improvements, the SC strategy incurs a significant overhead proportional to the number of sampled outputs. Taking MATH dataset as an example, evaluating the entire test set with SC (sampling size as 64 following Lewkowycz et al. (2022)) costs about 2000\$ through GPT-4 API, which is a significant burden for many researchers and organizations. Therefore, it is essential to minimize the cost of SC while maintaining performance.

The process of generating multiple samples in SC can be viewed as approximating the true answer distribution predicted by the language model. Then the most frequent one is taken as the final answer to mitigate the stochasticity of the single-sampling strategy. However, given that only the most confident answer is needed for SC, it is not necessary whether the whole answer distribution fits perfectly. Therefore, we argue that it is not necessary to directly generate all reasoning paths aligning with the preset sampling size for every input. Instead, the generation process can be serialized as smaller parts, each of which is named as sampling window. Considering both the small window and the large number of sampling outputs are generated from the same predicted answer distribution,

---

[†]Equal contributions.

[‡]Corresponding author.

[*]Our code and data have been released on `https://github.com/Yiwei98/ESC`.

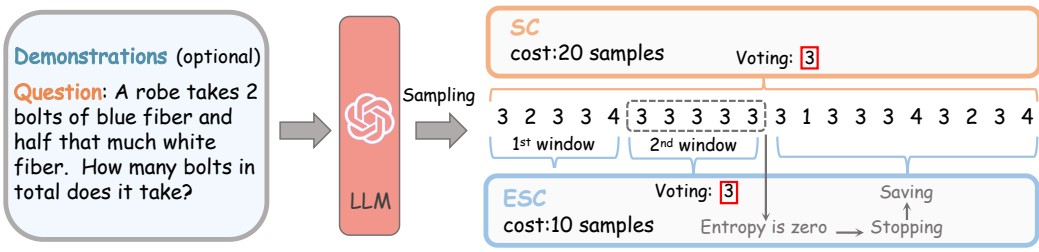

Figure 1: Full process of ESC compared with original SC. We divide the large sample size into several sequential small windows. Stop sampling when answers within a window are all the same, i.e., the entropy score of predicted answer distribution is zero.

it can be deemed as a probe to reveal some information of the true distribution with only a small sampling number.

For the answer distribution, one conjecture is that the correct ones are often concentrated, and the incorrect answers are scattered. We employ entropy as a representation of the answer distribution shape. Figure 2 shows the mean entropy value of correct and incorrect voting answer within a window respectively, showing that distributions with correct one as highest probability answer typically have much lower entropy values. Thus, it can be a indicator to determine whether sampling should continue. Based on this, we propose early-stopping self-consistency (ESC), truncating the sampling process with low entropy window. Figure 1 illustrates its process with an example. In order to maintain the performance as much as possible, we set the strictest threshold: the entropy equals zero, i.e., all generated samples within a window have the same answer. Stop sampling when this situation occurs to reduce sampling consumption while minimize performance impact.

Early stopping (Yao et al., 2007) is a widely used technique to prevent poor generalization when training models. But in this paper, we introduce this strategy to early stop the generation process for saving the cost. As with original SC, ESC is unsupervised and model-agnostic, without any human annotation and additional training. We derive the theoretical upper bound of inconsistent probability of the results with or without early stopping scheme in SC, indicating that ESC is highly likely to maintain performance. In addition, one control scheme for ESC is further derived to dynamically choose the performance-cost balance for different tasks and models by selecting the size of window and maximum sampling times for meeting the practice requirements.

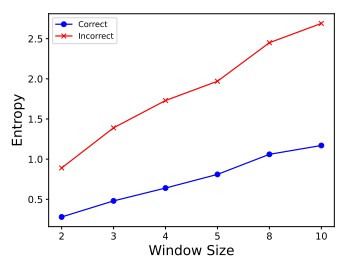

Figure 2: The mean entropy score within the window on MATH dataset from GPT-4.

We evaluate ESC on a wide range of arithmetic, commonsense and symbolic reasoning tasks over three language models with varying scales: GPT-4, GPT-3.5-Turbo and Llama-2 7b. The empirical results show that ESC reduces the sampling times by a significant margin on six popular benchmarks, including MATH (Hendrycks et al., 2021) (-33.8%), GSM8K (Cobbe et al., 2021) (-80.1%), StrategyQA (Geva et al., 2021) (-76.8%), CommonsenseQA (Talmor et al., 2019) (-78.5%), Coin Flip (-84.2%) and Last Letters (Wei et al., 2022) (-67.4%) , while attaining comparable performances. In additional experiments, we show our control scheme for ESC can predict the performance-cost balance accurately across various tasks and models, showcasing reliable application prospects. We also show ESC can robustly save cost considering different decoding settings and prompts.

## 2 METHOD

### 2.1 REVISITING SELF-CONSISTENCY

Self-consistency (Wang et al., 2023) capitalizes on the notion that a intricate problem requiring logical thinking usually admits several distinct approaches that all lead to the same correct answer. Based on this, multiple candidate predictions $\{\hat{y}^l\}^L$ to problem $x$ are suggested to generate through

sampling, and the most consistent $\hat{y}$ is selected as the final prediction through a voting process:

$$\hat{y} = \arg\max_i \sum_{l=1}^{L} \mathbf{1}_{\hat{y}^l = \mathrm{i}} \tag{1}$$

where $\mathbf{1}_{\hat{y}^l = \mathrm{i}}$ is the indicator function (equal to 1 when $\hat{y}^l$ is equal to prediction $i$, and 0 otherwise). We conduct a more in-depth derivation as follows:

$$\hat{y} = \arg\max_i \frac{\sum_{l=1}^{L} \mathbf{1}_{\hat{y}=\mathrm{i}}}{L} = \arg\max_i f^L(i) \tag{2}$$

where $f^L(i)$ denotes the frequency of the model's predicted outcome being $i$ in $L$ sampling instances. According to the Law of Large Numbers (Papoulis, 1990), as the sample size $L$ approaches infinity, $f^L(i) = P(i)$, where $P(i)$ represents the true probability of the model predicting the outcome as $i$. On this basis, we can further deduce:

$$\hat{y} = \arg\max_i P(i) \ \ (Given \ L \to \infty) \tag{3}$$

According to equation 3, we can re-conceptualize SC as the process of mitigating noise introduced by individual sampling through multiple samplings. Its objective is to ensure that the prediction with the highest probability, denoted as $\arg\max_i P(i)$ is chosen as the final answer.

From this perspective, we contend that when the entropy of $P$ is low (an extreme case being the one-hot-like distribution), a smaller value of $L$ ($L = 1$ is enough for one-hot-like distribution) can mitigate the impact of sampling noise according to the Law of Large Numbers.

## 2.2 Early-Stopping Self-Consistency

Building upon the aforementioned analysis, we propose **E**arly-**S**topping **S**elf-**C**onsistency (ESC) to achieve comparable performance to SC at a much lower sampling cost by adaptively adjusting the sampling times. The execution flow of ESC is illustrated in Algorithm 1.

---

**Algorithm 1** Early-Stopping Self-Consistency.

---

**Require:** model $\mathcal{M}$, dataset $\mathbb{D} = \{(x, y)\}^N$, window size $w$, max sampling size $L$, past sampling set $\mathbb{S}_{past}$
**Ensure:** predictions set $\mathbb{S}_{predictions}$

   $\mathbb{S}_{predictions} \leftarrow \varnothing$
   **for** $i \in$ range $N$ **do**
      $\mathbb{S}_{candidates} \leftarrow \varnothing, \mathbb{S}_{window} \leftarrow \mathbb{S}_{past}^i$ **if** $\mathbb{S}_{past} \neq \varnothing$ **else** $\varnothing$
      **for** $j \in$ range $(L//w)$ **do**
         $\mathbb{S}_{window} \leftarrow$ Sampling predictions $w$ times from $\mathcal{M}$ given $x^i$ **if** $\mathbb{S}_{window} = \varnothing$ **else** $\mathbb{S}_{window}$
         **if** predictions in $\mathbb{S}_{window}$ are the same **then**
            $\mathbb{S}_{candidates} \leftarrow \mathbb{S}_{window}$ **break**
         **end if**
         $\mathbb{S}_{candidates} \leftarrow \mathbb{S}_{candidates} + \mathbb{S}_{window}, \mathbb{S}_{window} \leftarrow \varnothing$
      **end for**
      $\mathbb{S}_{predictions} \leftarrow \mathbb{S}_{predictions}.\text{Append}(\arg\max_i \sum_{p \in \mathbb{S}_{candidates}} \mathbf{1}_{p=i})$
   **end for**
   **return** $\mathbb{S}_{predictions}$

---

We view a consecutive set of $w$ sampling predictions as an observation window for the probability distribution $P$ predicted by model $\mathcal{M}$ for input $x$. When these $w$ predictions are all the same, the entropy of $P$ is likely to be sufficiently low. At this point, the voting result of these $w$ samples is very likely to be equal to $\arg\max_i P(i)$, which is exactly the voting result that SC aims for. Therefore, we stop sampling when this situation occurs to save additional sampling consumption while barely affecting performance. We will iterate to obtain multiple observation windows until the preset sampling size $L$ is reached if no observation window meets the need during this procedure. The predictions of all the samples will constitute the final outcome according to equation 1.

## 2.3 Theoretical Analysis

Now, we analyze to what extent ESC will effect the performance compared with SC. For simplicity, we study the case where $L$ is infinite (we will examine the case of bounded $L$ in section 2.4.2). We

conduct a *one proportion z-test* with the following null hypothesis,

$$H_0 : \text{Prediction } p \text{ appears in an observation window } T \text{ times, where } p \neq \arg\max_i P(i).$$

According to the definition of *one proportion z-test*, the calculation formula for the z-statistic is:

$$z = \frac{(\hat{T} - T_\mu) * \sqrt{n}}{T_\sigma} \tag{4}$$

where $\hat{T}$ is the observed mean, $T_\mu$ represents the expected mean, $T_\sigma$ represents the expected standard deviation, and $n$ represents the observed times. Considering the null hypothesis $H_0$, where $p \neq \arg\max_i P(i)$, we can derive as follows:

$$P(p) \leq (P(p) + max(P))/2 \leq \sum_i P(i)/2 = 1/2 \tag{5}$$

As $w$ samples make up an observation window, according to Bernoulli distribution and equation 5:

$$T_\mu = w * P(p) \leq w/2$$
$$T_\sigma = \sqrt{w * P(p)(1 - P(p))} \leq \sqrt{w}/2 \tag{6}$$

Taking equation 6 into equation 4, z-statistic has the following lower bound when $\hat{T} \geq T_\mu$:

$$z \geq \frac{(\hat{T} - w/2) * \sqrt{n}}{\sqrt{w}/2} \tag{7}$$

The voting results of ESC are inconsistent with the voting results of SC if and only if $T$ equals to $w$ (when early-stop happens while $p \neq \arg\max_i P(i)$). When this situation occurs, we have the observed times $n = 1$ and $\hat{T} = w$, which we take into equation 7:

$$z \geq \frac{(w - w/2) * \sqrt{1}}{\sqrt{w}/2} = \sqrt{w} \tag{8}$$

Suppose $w = 8$, and we choose to reject $H_0$ if $z \geq \sqrt{8}$ (the lower bound in equation 8). In this case, the probability of a false positive is $\leq 2 \times 10^{-3}$ (one-sided p-value corresponding to $z = \sqrt{8}$). This means that when we conduct ESC with $w = 8$, the probability of the voting outcome being inconsistent with the voting outcome of SC (false positive) is $\leq 2 \times 10^{-3}$, which is extremely low. According to equation 8, we can see that as $w$ increases, the lower bound of $z$ increases accordingly, resulting in a smaller upper bound of the probability that ESC performance being affected.

## 2.4 CONTROL SCHEME FOR EARLY-STOP SELF-CONSISTENCY

In practical applications, the desired scenario is that we can adjust the ESC strategy (window size $w$, max sampling size $L$) based on our sampling budgets and performance requirements. Therefore, we propose a control scheme for ESC to achieve this goal. Specifically, we will deduce the expectation of voting performance and sampling cost under different $(w, L)$ settings based on the first observation window (denote its window size as $w_0$).

### 2.4.1 THE EXPECTATION OF SAMPLING COST

We first use the sampling frequency in the first observation window to approximate the true probability distribution $P$, and denote it as $\hat{P}$. Based on this, the probability of stopping sampling in each observation window (where the sampled values within the window are all the same) is:

$$\hat{P}_{stop} = \sum_i \text{pow}(\hat{P}(i), w) \tag{9}$$

According to Algorithm 1, the expected average sampling times $\hat{L}$ of dataset $\mathbb{D}$ is:

$$\mathbb{E}(\hat{L}) = \mathbb{E}_{\hat{P} \in \mathcal{M}(\mathbb{D})} \sum_{j=0}^{L//w-1} [(\hat{P}_{stop} \times \text{pow}(1 - \hat{P}_{stop}, j) \times j \times w) + \text{pow}(1 - \hat{P}_{stop}, L//w) \times L] + w_0 \tag{10}$$

see Appendix A.1 for detailed derivation. According to equation 10, we can determine the mapping relationship from the choice of $(w, L)$ to the expected sampling cost $\hat{L}$ based on the sampling results of the first observation window.

### 2.4.2 The Expectation of Voting Performance

When $L$ is bounded (practical scenarios), we discuss the probability of ESC voting outcome being inconsistent with $\arg\max_i P(i)$ with (denoted as $Q_w(\hat{P})$) and without (denoted as $Q_o(\hat{P})$) the occurrence of early-stop.

When early-stop happens, similar to the derivations in section 2.3, we substitute $\hat{P}$ to $P$ to calculate the z-statistic under the $\hat{P}$ distribution. We then calculate the sum of the probabilities that the voting outcome of ESC being inconsistent with the voting outcome of SC for all candidate $p$ when early-stop happens, where $p \neq \arg\max_i \hat{P}(i)$:

$$Q_w(\hat{P}) = \sum_{p \neq \arg\max_i \hat{P}(i)} \text{querying}(\frac{w - w * \hat{P}(p)}{\sqrt{w * \hat{P}(p)(1 - \hat{P}(p))}}) \tag{11}$$

where $\text{querying}(\cdot)$ represents the process of querying p-value corresponding to the z-statistic.

When early-stop not happens, ESC degenerates into SC. We view the whole $L$ samples as an observation window and has the following derivation of z-statistic:

$$z = \frac{\hat{T} - L * \hat{P}(p)}{\sqrt{L * \hat{P}(p)(1 - \hat{P}(p))}} \tag{12}$$

If prediction $p$ is selected as the voting outcome, then the the following inequality should hold: $\hat{T} \geq L/2$. Combining with equation 12, we derive the upper bound of $Q_o(\hat{P})$:

$$Q_o(\hat{P}) \leq \sum_{p \neq \arg\max_i \hat{P}(i)} \text{querying}(\frac{L/2 - L * \hat{P}(p)}{\sqrt{L * \hat{P}(p)(1 - \hat{P}(p))}}) \tag{13}$$

According to equation 9, the probability of that early-stop not happens is $\text{pow}(1 - \hat{P}_{stop}, L//w)$. Thus, the upper bound of the expected probability of ESC voting outcome being inconsistent with $\arg\max_i P(i)$ is:

$$\mathbb{E}(Q) \leq \mathbb{E}_{\hat{P} \in \mathcal{M}(\mathbb{D})}(1 - \text{pow}(1 - \hat{P}_{stop}, L//w)) \times Q_w(\hat{P}) + \text{pow}(1 - \hat{P}_{stop}, L//w) \times Q_o(\hat{P}) \tag{14}$$

### 2.4.3 Controllable Early-Stop Self-Consistency

Based on the above derivation, we propose our control scheme for ESC as shown in Algorithm 2. First, we sample $w_0$ times on the whole dataset. Based on the results of the first observation window, we calculate the expected sampling cost and performance under different settings of $(w, L)$. Finally, considering the sampling budget and performance requirements, we choose appropriate values of $(w, L)$ based on the respective expected values to execute ESC.

---

**Algorithm 2** Control Scheme for Early-Stop Self-Consistency.

---

**Require:** model $\mathcal{M}$, dataset $\mathbb{D} = \{(x, y)\}^N$, initial window size $w_0$ (recommended as 5), sampling budget $B$, performance expectation $P$.
**Ensure:** Predictions set $\mathbb{S}_{predictions}$
    $\mathbb{S}_{firstwindow} \leftarrow \varnothing$
    **for** $(x, y) \in \mathbb{D}$ **do**
        $\mathbb{S}_{window} \leftarrow$ Sampling predictions $w_0$ times from $\mathcal{M}$ given $x$
        $\mathbb{S}_{firstwindow} \leftarrow \mathbb{S}_{firstwindow} + \mathbb{S}_{window}$
    **end for**
    **for** $\forall (w, L)$ **do**
        $\mathbb{E}(\hat{L}) \leftarrow$ section 2.4.1 ($\mathbb{S}_{firstwindow}$, $w$, $L$), $\mathbb{E}(\hat{Q}) \leftarrow$ section 2.4.2 ($\mathbb{S}_{firstwindow}$, $w$, $L$)
        **if** $(\mathbb{E}(\hat{L}), \mathbb{E}(\hat{Q}))$ meets the need of $(B, P)$ **then**
            $\hat{w} \leftarrow w$, $\hat{L} \leftarrow L$ **break**
        **end if**
    **end for**
    **return** Algorithm 1($\mathcal{M}$, $\mathbb{D} = \{(x, y)\}^N$, $\hat{w}$, $\hat{L}$, $\mathbb{S}_{firstwindow}$)

---

# 3 EXPERIMENTS

## 3.1 EXPERIMENT SETUP

We evaluate the proposed ESC on six benchmark datasets from three categories of reasoning tasks: For arithmetic reasoning, we consider MATH (Hendrycks et al., 2021) and GSM8K (Cobbe et al., 2021). MultiArith (Roy & Roth, 2015), SVAMP (Patel et al., 2021), AddSub (Hosseini et al., 2014) and ASDiv (Miao et al., 2020) are not chosen in this paper because they are relatively simple. For commonsense reasoning, CommonsenseQA (Talmor et al., 2019) and StrategyQA (Geva et al., 2021) are used. For symbolic reasoning, we use Last Letter Concatenation and Coin Flip from Wei et al. (2022). The data version is from Kojima et al. (2022).

ESC is evaluated across three language models with varying scales: GPT-4 (OpenAI, 2023), GPT-3.5-Turbo and LLaMA-2 7B (Touvron et al., 2023). All experiments are conducted in the few-shot setting without training or fine-tuning the language models. To ensure a fair comparison, we use the same prompts as Wei et al. (2022). Details on the prompts used are given in Appendix.

The sampling temperature $T$ for MATH is 0.5 while for other datasets is 0.7. GPT-4 and GPT-3.5-Turbo samples predictions without truncating. For Llama 2, the threshold for top p truncation (Holtzman et al., 2020) is 0.9. Similarly to Wang et al. (2023), we provide an ablation study in Section 3.6 to show that ESC is generally robust to sampling strategies and parameters.

## 3.2 MAIN RESULTS

The baseline we compare to is chain-of-thought prompting with greedy decoding (CoT) and self-consistency (SC) with sampling. Following Lewkowycz et al. (2022), the sample size $L$ for MATH is 64 and for others is 40, and ESC uses the same value as maximum sample size. Accordingly, the window size $w$ for MATH is 8 and for others is 5. We report the results averaged over 10 runs and omit variance for limited space.

Table 1: Accuracy (%) and $\hat{L}$ (average actual number of generated samples in ESC, in gray) across six reasoning benchmarks. $\hat{L}$-SC denotes the accuracy of SC with sample size as $\hat{L}$.

|  |  | MATH | GSM8K | CSQA | SQA | Letter | Coinflip |
|---|---|---|---|---|---|---|---|
| GPT-4 | CoT | 50.44 | 87.70 | 83.71 | 78.63 | 93.12 | 100.00 |
|  | SC | 60.32 | 89.29 | 87.18 | 81.67 | 95.00 | / |
|  | ESC | 60.32 (0.00) | 89.29 (0.00) | 87.18 (0.00) | 81.70 (+0.03) | 94.98 (-0.02) | / |
|  | $\hat{L}$ | 42.40 (-21.60) | 7.98 (-32.02) | 9.29 (-30.71) | 7.19 (-31.39) | 6.32 (-33.68) | / |
|  | $\hat{L}$-SC | 59.98 (-0.34) | 89.07 (-0.22) | 86.49 (-0.69) | 81.40 (-0.27) | 94.59 (-0.39) | / |
| GPT-3.5 Turbo | CoT | 35.53 | 75.83 | 74.17 | 67.66 | 80.50 | 83.74 |
|  | SC | 49.97 | 85.69 | 78.10 | 75.90 | 83.21 | 99.54 |
|  | ESC | 49.96 (-0.01) | 85.67 (-0.02) | 78.10 (0.00) | 75.71 (-0.19) | 83.15 (-0.06) | 99.49 (-0.05) |
|  | $\hat{L}$ | 52.37 (-11.63) | 14.65 (-25.35) | 11.70 (-28.30) | 8.51 (-27.93) | 8.82 (-31.18) | 13.03 (-26.97) |
|  | $\hat{L}$-SC | 49.79 (-0.13) | 84.82 (-0.85) | 77.67 (-0.43) | 75.07 (-0.83) | 82.74 (-0.41) | 98.67 (-0.82) |
| Llama-2 7B | CoT | 5.09 | 18.07 | 65.28 | 46.23 | 14.87 | 54.74 |
|  | SC | 7.68 | 21.75 | 67.70 | 63.15 | 23.32 | 59.13 |
|  | ESC | 7.68 (0.00) | 21.74 (-0.01) | 67.68 (-0.02) | 63.01 (-0.14) | 23.32 (0.00) | 58.99 (-0.14) |
|  | $\hat{L}$ | 62.48 (-1.52) | 31.21 (-8.79) | 11.82 (-28.18) | 11.00 (-23.96) | 34.73 (-5.27) | 14.87 (-25.13) |
|  | $\hat{L}$-SC | 7.68 (0.00) | 21.52 (-0.22) | 66.97 (-0.71) | 61.19 (-1.96) | 23.11 (-0.21) | 58.11 (-0.88) |

**ESC significantly reduces costs while barely affecting performance.** Table 1 summarizes accuracy of CoT, SC, proposed ESC and $\hat{L}$, the average actual number of generated samples in ESC, for each dataset among three language models. The first observation is that SC outperforms CoT substantially, which confirms the effectiveness of the voting process for reasoning. For ESC, the $\hat{L}$ is largely smaller than the corresponding maximum sampling size $L$, while the accuracy has remained almost unchanged. Given that the accuracy of Coin Flip on GPT-4 is 100%, there is no need to conduct SC and ESC on it. We also test SC with $\hat{L}$ as the sampling size ($\hat{L}$-SC), whose accuracy

drops in accordance with the performance curve relative to the number of samples from Wang et al. (2023). Overall, ESC can reduce costs significantly while barely affecting performance. In other words, ESC can get higher accuracy under the same sampling costs.

Table 2: Reasoning accuracy (%) and $\hat{L}$ with various max sampling size $L$. The window size is 8.

| Model | Method | 16 | 24 | 32 | 40 | 48 | 64 |
|---|---|---|---|---|---|---|---|
| GPT-4 | SC | 58.92 | 59.40 | 59.77 | 59.95 | 60.07 | 60.31 |
| | ESC | 58.92 (0.00) | 59.40 (0.00) | 59.77 (0.00) | 59.95 (0.00) | 60.07 (0.00) | 60.31 (0.00) |
| | $\hat{L}$ | 13.56 (-2.44) | 18.72 (-5.28) | 23.67 (-8.33) | 28.49 (-11.51) | 33.21 (-14.79) | 42.41 (-21.59) |
| GPT-3.5 Turbo | SC | 47.34 | 48.48 | 49.02 | 49.40 | 49.65 | 49.96 |
| | ESC | 47.33 (-0.01) | 48.49 (+0.01) | 49.02 (0.00) | 49.41 (+0.01) | 49.64 (-0.01) | 49.96 (0.00) |
| | $\hat{L}$ | 14.84 (-1.16) | 21.38 (-2.62) | 27.76 (-4.24) | 34.02 (-5.98) | 40.20 (-7.80) | 52.37 (-11.63) |
| Llama-2 7B | SC | 7.10 | 7.28 | 7.40 | 7.45 | 7.54 | 7.70 |
| | ESC | 7.10 (0.00) | 7.28 (0.00) | 7.40 (0.00) | 7.45 (0.00) | 7.54 (0.00) | 7.70 (0.00) |
| | $\hat{L}$ | 15.88 (-0.12) | 23.72 (-0.28) | 31.52 (-0.48) | 39.29 (-0.71) | 47.04 (-0.96) | 62.48 (-1.52) |

**ESC is a scalable decoding process across sampling and window size.** We conduct experiments with various window size and sampling size to valid the scalability of ESC. Table 2 shows the performance across different maximum sampling sizes. First we can see the performance of SC continuously improves as sampling size $L$ increases, which is consistent with the results in (Wang et al., 2023). On this basis, ESC can significantly save costs while maintaining performance for different $L$. Figure 3 shows performance-cost balancing lines among three language models on GSM8K. ESC is robust to different window size and maximum sampling number. Please refer to Appendix-Figure 7 for results on other datasets.

**Cost savings are positively correlated with performance.** As shown in both Table 1 and Table 2, an obvious phenomenon is that cost savings are positively correlated with performance. It is intuitive since better performance often eliminates the need for larger sample size as recommended in Wang et al. (2023). However, ESC does not require any prior knowledge of model capabilities and task difficulty. In addition, a control scheme for ESC is also proposed and will be evaluated in Section 3.3.

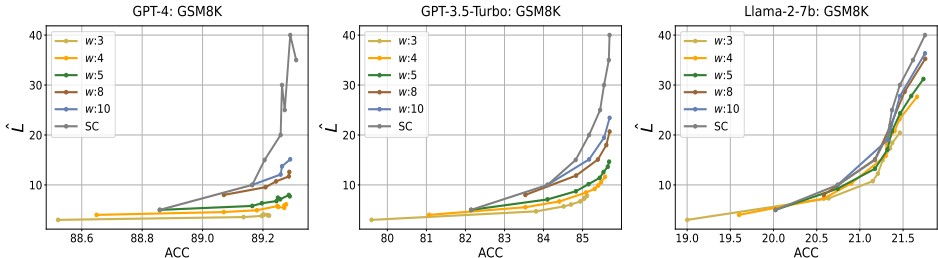

Figure 3: Robustness analysis of ESC regarding the observation window size $w$ and max sampling size $L$ on GSM8K with different models.

### 3.3 EFFECTIVENESS OF CONTROL SCHEME FOR ESC

To assess the effectiveness of control scheme for ESC, we compared the consistency between our predicted and actual values of sampling cost $\hat{L}$ and performance change percentage between SC and ESC $P_\delta$ on the GSM8K dataset. We chose L1 norm and Pearson correlation to measure the consistency.[*] As shown in Table 3, the Pearson correlations for both $\hat{L}$ and $P_\delta$ exceed 0.8, indicating a strong linear correlation between the predicted and actual values. Also, the L1 norm for both $\hat{L}$

---

[*]We vary the max sampling size to obtain multiple sets of results for calculating the Pearson correlation and mean L1 norm.

and $P_\delta$ are very low. These results indicate that the predictions we obtain based on equation 10 and equation 14 are highly reliable for balancing sampling cost and voting performance.

Table 3: Consistency between the predicted (through control scheme for ESC) and actual values of $\hat{L}$: sampling times, and $P_\delta$ (%): performance change percentage between SC and ESC. We choose L1 norm and Pearson correlation to measure the consistency. All the p-values $< 0.05$.

| Model | $\|\hat{L}^{act} - \hat{L}^{pre}\|_1$ | Pearson$(\hat{L}^{act}, \hat{L}^{pre})$ | $\|P_\delta^{act} - P_\delta^{pre}\|_1$ | Pearson$(P_\delta^{act}, P_\delta^{pre})$ |
|---|---|---|---|---|
| GPT-4 | 0.27 | 1.00 | 0.43 | 0.81 |
| GPT-3.5 Turbo | 0.62 | 1.00 | 0.06 | 0.86 |
| Llama-2 7B | 1.43 | 1.00 | 0.52 | 0.86 |

## 3.4 ESC FOR OPEN-ENDED GENERATIONS

Original SC is only suitable for problems that have fixed answers, while Jain et al. (2023) extended it for open-ended generation tasks by replacing voting through text similarity matching. We conduct ESC on MBPP dataset (Austin et al., 2021) with various sampling size (window size is 5). The results in Table 4 shows that ESC is also suitable for open-ended task.

Table 4: Reasoning accuracy (%) and $\hat{L}$ for MBPP with various max length L on GPT-3.5-Turbo.

| Method | 10 | 15 | 20 | 25 | 30 |
|---|---|---|---|---|---|
| SC | 61.96 | 61.96 | 62.04 | 62.15 | 62.18 |
| ESC | 61.96 (0.00) | 61.96 (0.00) | 62.02 (-0.02) | 62.11 (-0.04) | 62.15 (-0.03) |
| $\hat{L}$ | 5.62 (-4.38) | 6.02 (-8.98) | 6.32 (-13.68) | 6.57 (-18.43) | 6.79 (-23.21) |

## 3.5 INTERSECTION BETWEEN ESC AND SC

According to Section 2.3, the voting results of ESC should have a high probability of being consistent with SC. From Table 5 we can see that the intersection ratios of the voting results of SC and ESC are quite high, which indicates the upper bound of the performance being affected by ESC derived in Section 2.4.2 is reliable.

Table 5: Intersection ratio between ESC and SC.

| Model | GSM8K | SQA | Letter |
|---|---|---|---|
| GPT-4 | 99.69 | 99.69 | 99.96 |
| GPT-3.5-Turbo | 99.92 | 99.25 | 99.76 |
| Llama-2-7B | 99.77 | 99.13 | 99.75 |

## 3.6 ROBUSTNESS OF ESC

Additional experiments were conducted to further test the robustness of the proposed ESC, including its robustness to sampling parameters and prompts: (1) In Figure 4 (up) we show how ESC behaves for GSM8K as the decoding sampling temperature increases. Savings are consistent across different generation temperatures. (2) Figure 4 (bottom left corner) shows that ESC is robust to p values for top-p sampling. (3) Figure 4 (bottom right corner) indicates ESC can generalize to zero-shot manner. (4) Table 6 shows the accuracy of ESC and SC with different groups of demonstrations. We can see that ESC is robust to various demonstrations (see Appendix-Figure 6 for results on StrategyQA).

Table 6: Reasoning accuracy (%) and $\hat{L}$ for GSM8K on GPT-3.5-Turbo with different demonstrations. The max sampling size is 40 and window size is 5.

| Demonstration Groups | 1st | 2nd | 3rd | 4th | 5th |
|---|---|---|---|---|---|
| SC | 85.69 | 85.56 | 84.80 | 85.63 | 85.24 |
| ESC | 85.67 (-0.02) | 85.58 (0.02) | 84.80 (0.00) | 85.64 (0.01) | 85.23 (-0.01) |
| $\hat{L}$ | 14.65 (-25.35) | 14.53 (-25.47) | 15.64 (-24.36) | 14.76 (-25.24) | 14.59 (-25.41) |

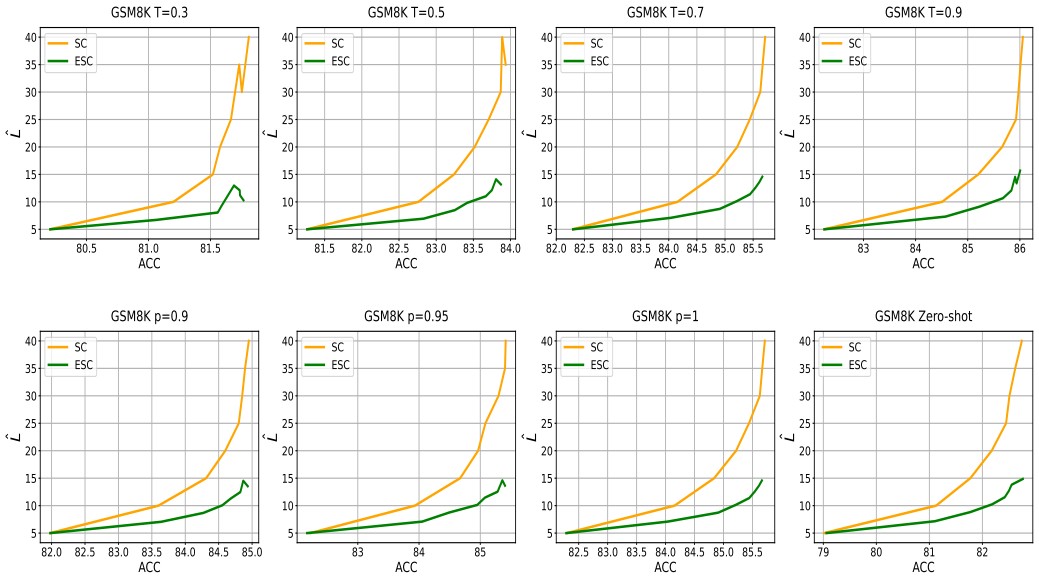

Figure 4: Robustness analysis of ESC regarding the sampling temperature $T$, $p$ in nucleus sampling, and zero-shot demonstration on GSM8K with GPT-3.5-Turbo.

## 4 RELATED WORK

**Chain-of-thought Reasoning**    Chain-of-thought prompting has been proven to be an effective method of solving complex reasoning problems (Wei et al., 2022). By following the pattern of gradually solving sub-problems, both few-shot CoT (Fu et al., 2023) and zero-shot CoT (Kojima et al., 2022) are capable of stimulating LLM reasoning abilities. On this basis, Least-to-most prompting (Zhou et al., 2023) suggests explicitly splitting the problem and solving them step by step. Zheng et al. (2023) iteratively generating answers and adding the historically generated answers as hints to the context to achieve the final convergence on the answer.

**Self Consistency**    Self-consistency (Wang et al., 2023) refers to a simple decoding strategy for further improving reasoning performance, leveraging the fact that complex reasoning tasks typically allow for more than one correct reasoning path. Jain et al. (2023) extend it for open-ended generation tasks like code generation and text summarization by replacing voting through text similarity matching. Li et al. (2023) assign appropriate weights for answer aggregation to achieve adaptive self-consistency. However, all of them require multiple sampling with the pre-set size, which will incur much more computation cost. Aggarwal et al. (2023) introduce an adaptive stopping criterion based on the amount of agreement between the samples so far, but it needs additional data to tune the hyperparameter and is sensitive to threshold. By contrast, ESC has no hyperparamter for stopping criterion and has a additional control scheme to meet the realistic requirements.

## 5 CONCLUSION

We have introduced a simple yet effective sampling process called early-stopping self-consistency (ESC). By stopping the decoding process with high confident window, ESC greatly reduce the cost of SC without sacrificing performance. A control scheme for ESC is further derivated to dynamically select the performance-cost balance for different tasks and models, which requires no extra prior knowledge of model capabilities and task difficulty. The empirical results show that ESC reduces the actual number of samples of chain-of-thought reasoning by a significant margin on six popular benchmarks, while attaining comparable performances. We also show control scheme for ESC can predict the performance-cost trade-off accurately across various tasks and models. The additional evaluations indicate that ESC can robustly save cost considering different decoding settings and prompts, and even on open-ended generation tasks.

ACKNOWLEDGMENTS

This work is supported by Beijing Natural Science Foundation (No.4222037, L181010).

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

# A APPENDIX

## A.1 DERIVATION OF THE EXPECTATION OF SAMPLING COST

Given the probability of stopping sampling in each observation window as $\hat{P}_{stop}$, the probability of continue sampling is $1 - \hat{P}_{stop}$. Therefore, the probability of early-stop occurring after $j$ windows is $\hat{P}_{stop} \times \text{pow}(1 - \hat{P}_{stop}, j)$. On this basis, the expected sampling cost in the first $L//w - 1$ windows can be denoted as:

$$\mathbb{E}(\hat{L_1}) = \mathbb{E}_{\hat{P} \in \mathcal{M}(\mathbb{D})} \sum_{j=0}^{L//w-1} (\hat{P}_{stop} \times \text{pow}(1 - \hat{P}_{stop}, j) \times j \times w) \tag{15}$$

If no early-stop occurs after observing $L//w - 1$ windows, then the maximum sampling size $L$ is reached upon the next observation window. The expected sampling cost of this part can be denoted as:

$$\mathbb{E}(\hat{L_2}) = \mathbb{E}_{\hat{P} \in \mathcal{M}(\mathbb{D})} \sum_{j=0}^{L//w-1} \text{pow}(1 - \hat{P}_{stop}, L//w) \times L \tag{16}$$

Including $w_0$, the overall expected sampling cost is:

$$\mathbb{E}(\hat{L}) = \mathbb{E}_{\hat{P} \in \mathcal{M}(\mathbb{D})} \sum_{j=0}^{L//w-1} [(\hat{P}_{stop} \times \text{pow}(1 - \hat{P}_{stop}, j) \times j \times w) + \text{pow}(1 - \hat{P}_{stop}, L//w) \times L] + w_0 \tag{17}$$

## A.2 COMPARISON WITH ADAPTIVE-CONSISTENCY (AC)

Aggarwal et al. (2023) introduce an adaptive stopping criterion based on the amount of agreement between the samples so far, but it needs additional data to tune the hyperparameter and is sensitive to threshold. But for proposed ESC, the stopping criterion needs no hyperparameter due to the most conservative strategy to maintain the performance, i.e., all the answers within a window are same. Thus ESC can be conducted directly for different tasks and models, without any validation set.

Another drawback of AC is generating samples step by step, which means each sample requires one input. Considering the demonstrations for in-context learning (usually 8 examples) have a lot of tokens, it will cost quite a portion of the budget. By contrast, ESC generates samples in multiple sampling windows, thus samples within one window can share the same input. Table 7 shows that ESC can get higher accuracy with less sampling cost comparing with AC.

Table 7: Prompt token count (tokens/item, denoted as #prompt), completion token count (tokens/item, denoted as # completion), average sampling cost ($/item) and accuracy (%) resulting from different methods. The best and second-best results are emphasized with bold and underline.

| Datasets | Method | GPT-3.5 Turbo | | | | GPT-4 | | | |
|---|---|---|---|---|---|---|---|---|---|
| | | # prompt | # completion | Cost | Acc | # prompt | # completion | Cost | Acc |
| GSM8K | SC | 496.9 | 2909.0 | 0.0084 | **85.69** | 495.9 | 2646.5 | 0.2316 | **89.29** |
| | AC | 4930.3 | 813.2 | 0.0087 | 85.66 | 2927.6 | 412.5 | 0.1501 | 89.28 |
| | ESC | 1469.2 | 1220.8 | **0.0052** | 85.67 | 793.0 | 564.8 | **0.0769** | 89.29 |
| CSQA | SC | 428.7 | 1382.9 | 0.0043 | **78.10** | 427.8 | 1474.6 | 0.1351 | **87.18** |
| | AC | 3522.3 | 294.2 | 0.0055 | 78.05 | 2844.5 | 271.5 | 0.1355 | 87.15 |
| | ESC | 1003.2 | 420.9 | **0.0024** | 78.10 | 794.6 | 384.0 | **0.0625** | 87.18 |
| Letter | SC | 166.0 | 1463.1 | 0.0041 | **83.21** | 165.0 | 1396.9 | 0.1184 | **95.00** |
| | AC | 1057.5 | 233.6 | 0.0020 | 83.15 | 791.7 | 167.5 | 0.0451 | 94.97 |
| | ESC | 293.0 | 323.7 | **0.0012** | 83.15 | 208.7 | 220.7 | **0.0260** | 94.98 |

### A.3 COMPARISONS BETWEEN SC AND ESC WHEN L IS RELATIVELY SMALL (10 AND 20)

we conduct experiments with smaller sampling sizes $L$. From the results in Table 8, we found that ESC can consistently save sampling costs in such scenarios, while achieving improvement compared to SC under the same costs.

Table 8: Reasoning accuracy (%) and $\hat{L}$ with various max sampling size $L$. The window size is 5.

| Model | Method | CSQA | | GSM8K | | Letter | |
|---|---|---|---|---|---|---|---|
| | | $L=10$ | $L=20$ | $L=10$ | $L=20$ | $L=10$ | $L=20$ |
| GPT-4 | SC | 86.49 | 86.83 | 89.16 | 89.24 | 94.66 | 94.79 |
| | ESC | 86.49 (0.00) | 86.84 (+0.01) | 89.16 (0.00) | 89.24 (0.00) | 94.66 (0.00) | 94.80 (+0.01) |
| | $\hat{L}$ | 5.92 (-4.08) | 7.27 (-12.73) | 5.79 (-4.21) | 6.76 (-13.24) | 5.57 (-4.43) | 5.92 (-14.08) |
| | $\hat{L}$-SC | 86.07 (-0.42) | 86.31 (-0.52) | 88.95 (-0.21) | 89.02 (-0.22) | 94.58 (-0.08) | 94.58 (-0.21) |
| GPT-3.5 Turbo | SC | 77.63 | 77.93 | 84.10 | 85.15 | 82.80 | 83.00 |
| | ESC | 77.63 (0.00) | 77.91 (-0.02) | 84.10 (0.00) | 85.15 (0.00) | 82.81 (+0.01) | 82.98 (-0.02) |
| | $\hat{L}$ | 6.47 (-3.53) | 8.60 (-11.40) | 7.09 (-2.91) | 10.14 (-9.86) | 6.14 (-3.86) | 7.36 (-12.64) |
| | $\hat{L}$-SC | 77.39 (-0.24) | 77.55 (-0.38) | 83.31 (-0.79) | 84.10 (-1.05) | 82.45 (-0.35) | 82.65 (-0.35) |
| Llama2-7b | SC | 66.89 | 67.25 | 20.74 | 21.32 | 21.21 | 22.49 |
| | ESC | 66.89 (0.00) | 67.26 (+0.01) | 20.74 (0.00) | 21.32 (0.00) | 21.21 (0.00) | 22.48 (-0.01) |
| | $\hat{L}$ | 6.59 (-3.41) | 8.78 (-11.22) | 8.99 (-1.01) | 17.06 (-2.94) | 9.67 (-0.33) | 18.47 (-1.53) |
| | $\hat{L}$-SC | 66.80 (-0.09) | 66.84 (-0.41) | 20.59 (-0.15) | 21.21 (-0.11) | 21.21 (-0.00) | 22.32 (-0.17) |

### A.4 DISCUSSION ON THE CHOICE OF INTRODUCING OBSERVATION WINDOW FOR THE DESIGN OF EARLY-STOPPING STRATEGY

We design the stopping strategy with the introduction of window for the following two reasons. Firstly, we break the sampling process only if all of the samples in the latest window are consistent, thus avoiding any hyper-parameter. If sample one by one and stop based on the observation of the sampled samples, obviously we cannot adopt such a strict truncation condition. In this case, we need to introduce a certain statistic and its corresponding threshold (hyper-parameter), which is hard to be determined in prior. Secondly, we have actually considered using the normalized entropy of the sampled samples as the statistical value for cut-off. As shown in Figure 5 , we found that this method not only has the hyper-parameter problem mentioned above, but also has no advantage over ESC in terms of performance-cost trade-off. We believe this is because examining the cut-off point after each single sampling is too frequent, introducing greater randomness. This makes the model more likely to early-stop without sufficient sampling.

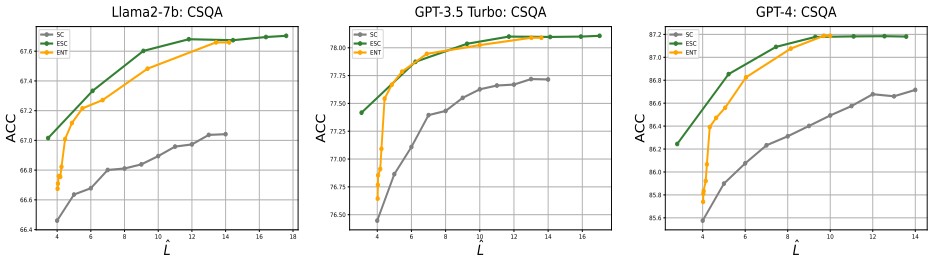

Figure 5: Comparisons between SC, ESC and ENT(using entropy to determine the cut-off point) on CSQA with different models.

### A.5 THE COMPARISONS BETWEEN ESC AND OTHER STRONG REASONING BASELINES.

ESC achieves better results compared to SC with the same overhead by saving sampling costs while almost not sacrificing performance. From the perspective of performance improvement, we are curious about the comparison results between ESC and advanced reasoning methods that focus on per-

formance enhancement. For this purpose, we compared ESC with Progressive-Hint Prompting(PHP) (Zheng et al., 2023) in terms of both performance and overhead. PHP improves performance by approaching answers through the stepwise generation of clues. As shown in Table 9, we found that ESC outperformed CoT by 9.84%, higher than PHP's 9.25%. Additionally, ESC incurs less sampling overhead. PHP, due to the need for multiple inputs to the model with previously enhanced prompts, has a larger overhead in the # prompt part. Overall, ESC achieves better performance with lower sampling costs.

Table 9: Prompt token count (tokens/item, denoted as #prompt), completion token count (tokens/item, denoted as # completion), average sampling cost ($/item) and accuracy (%) comparison between CoT, PHP (Zheng et al., 2023) and ESC on GSM8K with GPT-3.5-Turbo. Max sampling size of ESC is 40. CoT and PHP apply greedy search as sampling strategy.

| Method | # prompt | # completion | Sampling Cost | Accuracy |
|--------|----------|--------------|---------------|----------|
| CoT | 496.9 | 72.7 | 0.0006 | 75.83 |
| PHP | 6552.0 | 360.8 | 0.0072 | 85.08 |
| ESC | 1469.2 | 1220.8 | 0.0052 | 85.67 |

## A.6 THE ORTHOGONALITY OF ESC AND OTHER STRONG REASONING BASELINE.

To assess the orthogonality of ESC with other advanced reasoning methods, we applied ESC on PHP. As shown in Table 10, PHP w. ESC achieved similar performance with PHP w. SC while significantly reduced sampling overhead. This indicates that existing advanced methods can significantly reduce costs by applying ESC with almost no sacrifice in performance.

Table 10: Reasoning accuracy (%) and $\hat{L}$ for GSM8K on GPT-3.5-Turbo with Progressive-Hint Prompting(PHP) (Zheng et al., 2023). The window size is 5.

| Max sampling size | 10 | 20 | 30 | 40 |
|-------------------|-----|-----|-----|-----|
| PHP w. SC | 86.32 | 86.64 | 86.76 | 87.00 |
| PHP w. ESC | 86.32 (0.00) | 86.62 (-0.02) | 86.77 (+0.01) | 86.98 (-0.02) |
| $\hat{L}$ | 6.15 (-3.85) | 7.83 (-12.17) | 9.15 (-20.85) | 10.26 (-29.74) |
| $\hat{L}$-PHP w. SC | 86.02 (-0.30) | 86.29 (-0.35) | 86.32 (-0.44) | 86.35 (-0.65) |

## A.7 ADDITIONAL RESULTS

Figure 6 shows the robustness of ESC regarding the sampling temperature $T$, $p$ in nucleus sampling, and zero-shot demonstration on StrategyQA with GPT-3.5-Turbo. Figure 7 shows the robustness of ESC regarding the observation window size $w$ and max sampling size $L$ on multiple datasets with different models.

## A.8 PROMPTING DETAILS

We list the details of the prompts used for MATH dataset. Following Wang et al. (2023), we use the same prompts for other datasets as in Wei et al. (2022) for fair comparison.

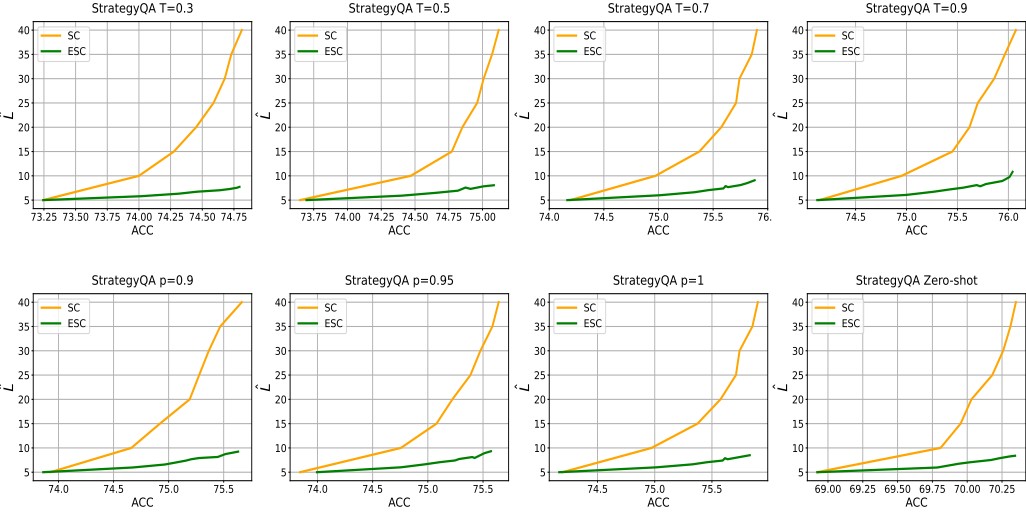

Figure 6: Robustness analysis of ESC regarding the sampling temperature $T$, $p$ in nucleus sampling, and zero-shot demonstration on StrategyQA with GPT-3.5-Turbo.

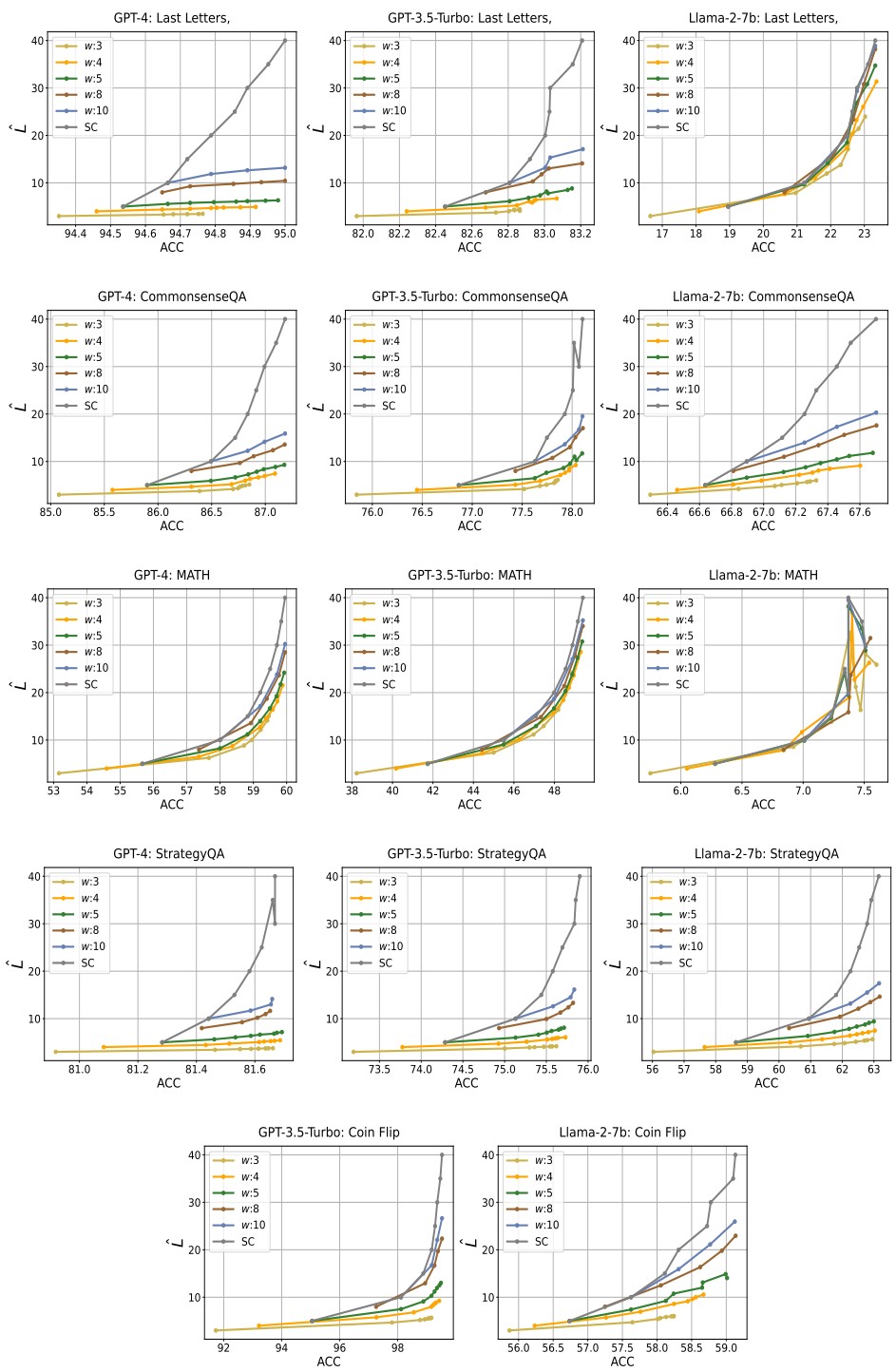

Figure 7: Robustness analysis of ESC regarding the observation window size $w$ and max sampling size $L$ on multiple datasets with different models.

Table 11: Few-shot prompts for MATH dataset.

**system:** Think the question step by step and give the answer.

**user:** Problem: Find the domain of the expression $\frac{\sqrt{x-2}}{\sqrt{5-x}}$.

**assistant:** Solution: The expressions inside each square root must be non-negative. Therefore, $x - 2 \geq 0$, so $x \geq 2$, and $5 - x \geq 0$, so $x \leq 5$. Also, the denominator cannot be equal to zero, so $5 - x > 0$, which gives $x < 5$. Therefore, the domain of the expression is $\boxed{[2, 5)}$.

**user:** Problem: Compute: $55 \times 1212 - 15 \times 1212$ .

**assistant:** Solution: We have $55 \times 1212 - 15 \times 1212 = 1212(55 - 15) = 1212(40) = 4848(10) = \boxed{48480}$.

**user:** Problem: Compute $\binom{16}{15}$.

**assistant:** Solution: $\binom{16}{15} = \binom{16}{1} = \boxed{16}$.

**user:** Problem: Find $x$, where $x$ is a square number which is divisible by four, and between 39 and 80.

**assistant:** Solution: We know that $x$ is between 39 and 80, and since $6^2 = 36 < 39$ and $9^2 = 81 > 80$, this means that $6^2 < x < 9^2$. This leaves us with two possibilities for $x$, which are $7^2 = 49$, and $8^2 = 64$. We then see that only 64 is divisible by four, so $x = \boxed{64}$.

**user:** Problem: Solve the inequality

$$\frac{(x-2)(x-3)(x-4)}{(x-1)(x-5)(x-6)} > 0.$$

**assistant:** Solution: We can build a sign chart, but since all of the factors are linear, we can track what happens to the expression as $x$ increases. At $x = 0$, the expression is positive. As $x$ increases past 1, the expression becomes negative. As $x$ increases past 2, the expression becomes positive, and so on. Thus, the solution is

$$x \in \boxed{(-\infty, 1) \cup (2, 3) \cup (4, 5) \cup (6, \infty)}.$$

**user:** Problem: A right circular cone has a volume of $12\pi$ cubic centimeters. The height of the cone is 4 cm. How many centimeters is the circumference of the base of the cone, in terms of $\pi$?

**assistant:** Solution: The volume of a cone is $\frac{1}{3}\pi r^2 h$. We are given that the volume is $12\pi$ and the height is 4. Thus, $\frac{1}{3}\pi r^2 \cdot 4 = 12\pi$. Solving for $r$, we find $r = 3$. Therefore, the circumference of the base is $2\pi r = \boxed{6\pi}$.

**user:** Problem: How many perfect squares less than 1000 have a ones digit of 2, 3 or 4?

**assistant:** Solution: Checking the squares from $1^2$ to $10^2$, we see that no squares end in 2 or 3, while a square ends in 4 if its square root ends in 2 or 8. Since $31^2 < 1000 < 32^2$, we see that the squares less than 1000 ending in 4 are $2, 8, 12, 18, 22, 28$. Thus the desired answer is $\boxed{6}$.

**user:** Problem: The diagonals of a rhombus are 10 inches and 24 inches. What is the perimeter of the rhombus, in inches?

**assistant:** Solution: The diagonals are perpendicular bisectors of each other, so therefore the side length of the rhombus can be calculated as $\sqrt{5^2 + 12^2} = 13$. Therefore, the perimeter of the rhombus is $4 \times 13 = \boxed{52}$ inches.

