# OpenReview forum: "Escape Sky-high Cost: Early-stopping Self-Consistency for Multi-step Reasoning"
_ICLR.cc/2024/Conference — ICLR 2024 poster_

### Official Review · Reviewer_UrCF · 2023-10-27

**Soundness:** 3 good
**Presentation:** 3 good
**Contribution:** 3 good
**Rating:** 8
**Confidence:** 4

**Summary:**

To address the issue of high costs associated with self-consistency (SC), this paper introduces an approach called Early-Stop Self-Consistency (ESC). ESC incorporates an early-stop strategy into SC to reduce the overall number of samples needed. The method achieves this by dividing the large sample size used in SC into smaller sequential windows, and it stops sampling when all answers within a window are the same. Additionally, the paper presents a control scheme for ESC that dynamically selecting the size of window and maximum sampling times for different tasks and models. The effectiveness and reliability of ESC are supported by solid theoretical guarantee and extensive experiments. The empirical results demonstrate that ESC significantly reduces the average number of samples required in SC across six benchmark tasks, all while maintaining comparable performance levels.

Contributions:
(1) This paper introduces an early-stop self-consistency method (ESC) to significantly reduce the computational cost of self-consistency while maintaining comparable performance.
(2) This paper also puts forth a control scheme for ESC that assists in the selection of an optimal window size and maximum sampling times, considering the sampling budget and performance requirements.
(3) Furthermore, this paper offers ample theoretical evidence to uphold the reliability of ESC.

**Strengths:**

(1) The method is simple and effective.

(2) It is backed by a solid theoretical foundation.

(3) Extensive experiments have been conducted to confirm its effectiveness and reliability.

**Weaknesses:**

(1) A related paper with a similar idea, called "Let’s Sample Step by Step: Adaptive-Consistency for Efficient Reasoning with LLMs" (https://arxiv.org/pdf/2305.11860.pdf), was not referenced.

(2) In Table 1, there appear to be inaccuracies in some of the results highlighted in green. For instance, in the row labeled "Lˆ-SC (GPT4)" and the column labeled "SQA," the value "(-0.27)" should actually be "(+0.87)" because the correct difference is 0.78 (81.42 - 80.55 = 0.78). Similar issues can be found in the "SQA" column. Additionally, it's puzzling that in the "SQA" dataset, Lˆ-SC outperforms SC, even though SC has a larger sample size. This phenomenon requires further explanation.

**Questions:**

Question:

(1) In the "SQA" dataset, Lˆ-SC outperforms both SC and ESC, even though SC has a larger sample size. If there are no data errors, is there any possible reasonable explanation?

Suggestion:

(1) In Table 1, the accuracy of Lˆ-SC seems decreases slightly (less than 0.5%) in more than half situations. Therefore, it might not be accurate to claim "a large margin", as the paper does, that "We also test SC with Lˆ as the sampling size (Lˆ-SC), whose accuracy drops by a large margin.".

---

> ### Author Response · Authors · 2023-11-18
> **Response to reviewer UrCF**
>
> Thank you for the positive and detailed review! We are very encouraged that you found our method to be simple, effective and robust. Below we address your comments.
>
> **Comparison with Adaptive-Consistency**
>
> Thanks for reminding and we will add citations to [1] and add more discussions on the novelty of our method:
>
> (1) Having additional control scheme
>
> Besides basic early-stopping self-consistency (ESC), we also propose its control scheme to dynamically choose the performance-cost balance. As we all know, along with the number of samples increases, performance will increase as well. For the realistic requirements, it is more practical to choose the affordable sample size rather than just try to maximize performance. We have derived the expectation of $\hat{L}$ (sampling number) and the upper bound of $\mathbb{E}(Q)$ (inconsistent probability). The experiments in Section 3.3 have shown that the predictions we obtain based on Equation 10 and Equation 14 are highly reliable for balancing sampling cost and voting performance.
>
> (2) Less sampling cost
>
> Adaptive-consistency (AC) generates samples step by step, which means each sample requires one input. Considering the demonstrations for in-context learning (usually 8 examples) have a lot of tokens, it will cost quite a portion of the budget. By contrast, ESC generates samples in multiple sampling windows, thus samples within one window can share the same input. Table 7 in updated paper shows that ESC can get higher accuracy with less sampling cost comparing with AC. A quick preview is below:
>
> | Method      |    \# prompt | \# completion  | Cost | Acc|
> | :--------: | :--------:| :--: | :--: | :--: |
> | SC  | 496.9 |  2909.0   | 0.0084 | 85.69|
> | AC     |   4930.3 |  813.2  | 0.0087|85.66|
> | ESC      |    1469.2 | 1220.8  | 0.0052|85.67|
>
> (3) No hyperparameters
>
> It is necessary for AC to tune the hyperparameter, specifically the confidence threshold. But for proposed ESC, the stopping criterion needs no hyperparameter due to the most conservative strategy to maintain the performance, i.e., all the answers within a window are same. Thus ESC can be conducted directly for different tasks and models, without any validation set. Besides, section 3.2 has proved the robustness regrading to window size w and max sampling size L.
>
> (4) More solid theoretical guarantees
>
> We provide more solid theoretical guarantees than AC lying in two parts: firstly, we rigorously derive the theoretical upper bound of the probability being inconsistent with the case where sampling size is infinite (true distribution of the model predictions according to Equation 3). While AC only considers the condition with next n samples, which is less rigorous; secondly, the upper bound of ESC is much more tight than AC. According to Equation 8, the bound value is only in the interval of $\leq 4 \times 10^{-2}$ to $\leq 2 \times 10^{-5}$ when window size varies from 3 to 10. By comparison，the confidence threshold of AC is chosen as 95\%.
>
> **Explanation and Fixing issues in Table 1**
>
> We apologize that we have filled in the inconsistent values about SQA. That is because the candidate answer for question in SQA is either 'True' or 'False', but there will be some cases that the model generates neither of them. So there are two ways to vote the final answer: one is regarding these noisy answers as another type, e.g. '-1', to be included for voting; the other is removing this type of answers and only voting 'True' or 'False'. Our first choice was to vote based on the former criterion, but later we realized that the latter one made more sense, but we have forgotten to correct some values. We have fixed this problem in the revision.
>
> **Modification of the description about Table 1**
>
> We thank you for your suggestion! We used the description "by a large margin" because we compared the performance degradation of L-SC with the change between SC and ESC. Sorry for misleading and we have modified the depiction.
> Here is the reason for this phenomenon: In the paper of vanilla self-consistency [2], the large improvements come from the comparison between single greedy sample and 40 samples. The author plotted the accuracy with respect to varying numbers of samples. Within a certain interval, the performance change is not significant either.
>
> [1] Aggarwal et al., Let's Sample Step by Step: Adaptive-Consistency for Efficient Reasoning with LLMs, EMNLP 2023.
>
> [2] Wang et al., Self-consistency improves chain of thought reasoning in language
> models, ICLR 2023.

---

> > ### Comment · Reviewer_UrCF · 2023-11-22
> >
> > Dear Authors,
> >
> > I appreciate your efforts in addressing the reviews.
> >
> > I consider this to be a good paper, and I maintain my score of 8 points.

---

### Official Review · Reviewer_rrBP · 2023-10-30

**Soundness:** 3 good
**Presentation:** 3 good
**Contribution:** 3 good
**Rating:** 5
**Confidence:** 4

**Summary:**

The paper presents a new technique called early-stopping self-consistency (ESC) aimed at improving the computational efficiency of machine learning models, particularly in the context of complex reasoning tasks. Leveraging the essence of Chain-of-thought Reasoning and Self Consistency, the authors introduce ESC as a mechanism to strike a balance between computational cost and performance. Through extensive experiments, the paper claims significant reduction in computational overhead without a noticeable drop in performance.

**Strengths:**

Originality: The introduction of ESC offers a fresh perspective in the realm of efficient machine learning algorithms.
Quality: The experimental setup, including testing on six benchmarks, demonstrates the thoroughness of the research.
Clarity: The paper, for the most part, is well-written and concepts are explained clearly.

**Weaknesses:**

Comparison with State-of-the-art: It would be helpful to see direct comparisons with current state-of-the-art methods in terms of efficiency and performance.
Generalizability: The paper could discuss potential limitations or scenarios where ESC might not be the optimal solution.

**Questions:**

The Early-Stop Consistency (ESC) strategy is an optimized or "pruned" version of the Self-Consistency (SC) method, and its effect is not improved compared to SC. From this point of view, the method is lack of novelty. The primary innovation of ESC lies not in a theoretical advance but in its practical utility. It addresses real-world constraints by optimizing the balance between computational expenditure and performance fidelity.

---

> ### Author Response · Authors · 2023-11-18
> **Response to reviewer rrBP**
>
> We thank your insightful comments! We understand your concerns, which are also very important to us. Below we clarify:
>
> **Comparison with state-of-the-art method**
>
> We thank you for this advice, which we believe helps strengthen our results.
> Per your suggestion, we conduct the experiment compared with Progressive-Hint Prompting (PHP) [1], a state-of-art method for arithmetic reasoning. In the updated paper, Table 9 from Appendix A.5, ESC (sampling size is 40) gets higher accuracy than PHP (greedy search) with less sampling cost. A quick preview is below:
>
> | Method      |    \# prompt | \# completion  | Sampling Cost | Accuracy|
> | :--------: | :--------:| :--: | :--: | :--: |
> | CoT  | 496.9 |  72.7   | 0.0006 | 75.83|
> | PHP     |   6552.0 |  360.8  | 0.0072|85.08|
> | ESC      |    1469.2 | 1220.8  | 0.0052|85.67|
>
> Actually, ESC is a plug-and-play technique that can be incorporated with most of state-of-art methods, given that self-consistency (SC) is usually orthogonal to them. So we think it will be more meaningful to verify its generalization ability to SOTA methods by combining them.
> Also taking PHP for example in Table 10, when applying our ESC to PHP, the sampling cost drops significantly while the accuracy is basically unchanged. We believe results from the study can expand the scope of the impact of ESC.
>
>
> |Max sampling size|10|20|30|40|
> |:-:|:-:|:-:|:--:|:-:|
> |PHP w. SC|86.32|86.64|86.76|87.00|
> |PHP w. ESC|86.32|86.62|86.77|86.98|
> |$\hat{L}$|6.15|7.83|9.15|10.26|
> |$\hat{L}$-PHP w. SC|86.02|86.29|86.32|86.35|
>
> **Discussion of limitations**
>
> According to Table 1, the decrease in sampling number is relatively small for Llama-2 model and MATH dataset, respectively. These two findings may reveal that the effect of ESC will be less significant under very difficult tasks and low-capability models.
> It makes sense since the lower accuracy will lead to fewer opportunities to reach the stop criterion, and need more samples to vote the right answer.
>
> **Novelty of ESC**
>
> Although ESC can not improve the performance compared with original SC under the same max sampling size, it can be achieved with same practical sampling cost. So from this perspective, ESC can also be seen as an method to improving effectiveness because it can get higher accuracy when enlarge the max sampling size to reach the same cost.
> Beside, as mentioned above, the ESC is also a plug-and-play method that can be incorporated with other competitive methods.
> Although ESC is more about practical significance, we provide solid theoretical guarantees to support it with rigorous upper bound, making it a key contribution in this field.
>
> [1] Zheng et al., Progressive-hint prompting improves reasoning in large language models, arXiv.2304.09797.

---

### Official Review · Reviewer_J879 · 2023-11-01

**Soundness:** 2 fair
**Presentation:** 2 fair
**Contribution:** 2 fair
**Rating:** 5
**Confidence:** 3

**Summary:**

The paper proposes to use an early stopping criterion, based on answer entropy, when sampling alternative answers from a LLM in a self-consistency (SC) setting. SC is a form of ensemble answering, where multiple answers are sampled and a vote decides on the final answer. With early stopping answers are sampled window-wise iteratively until the whole window contains the same answer. Experiments show that this can reduce the number of necessary calls to a LLM while maintaining similar accuracy in reasoning benchmarks.

**Strengths:**

LLMs are a popular topic currently and their execution is costly, either in monetary terms or computationally. Therefore, it is a good approach to reduce the number of calls necessary, as is proposed in the paper.
It is also a positive thing that existing proven techniques and statistical approaches are re-visited and used in these settings, such as early stopping or using answer entropy as a cut-off criterion.
The experimental evaluation confirms the suitability of the approach over the more exhaustive standard SC technique. Experiments are extensive and consider many facets of the proposed approach.

**Weaknesses:**

The contribution is not particularly strong. Early stopping or using the confidence respectively the variation in multiple answers in an ensemble of answers is a well known technique. While we have (maybe, I'm not sure) not seen this in LLM sampling, it is not a particularly strong contribution in the context of an ICLR paper.

I'm also not sure we actually need the notion of the window in the method or if other statistical measurements of the confidence resp. variability  could be used to determine the cut-off point. Unfortunately, this has not been discussed.

**Questions:**

No specific questions

---

> ### Author Response · Authors · 2023-11-18
> **Response to reviewer J879**
>
> We thank your valuable comments! We understand your concerns, which are also very important to us. Below we clarify:
>
> **Contribution of our work**
>
> Firstly, the motivation of ESC is quite straightforward and significant. Reducing inference costs is a crucial topic especially for LLMs. To this end, we propose a simple and effective method to considerably reduce the cost of SC without sacrificing performance.
> We conduct extensive experiments to confirm that ESC can robustly save cost considering different large models, tasks (even open-ended generation tasks), decoding settings, prompts, and combination with SOTA methods (e.g., PHP, in Appendix A.6).
> Although ESC is simple and intuitive, we provide solid theoretical guarantees for it, which means that ESC is not a naive and heuristic method but backed with rigorous upper bound.
> Beyond that, we further develop a control scheme to dynamically choose the performance-cost balance according to the sampling budget and performance requirements, which is also important for realistic applications.
>
> **Discussion on the choice of introducing observation window for the design of early-stopping strategy**
>
> We want to firstly clarify that using the consistency of samples within the window as cut-off point is also a strategy based on variability.
>
> We design the stopping strategy with the introduction of window for the following two reasons:
>
> (1) We break the sampling process only if all of the samples in the latest window are consistent, thus avoiding any hyper-parameter. If sample one by one and stop based on the observation of the sampled samples, obviously we cannot adopt such a strict truncation condition. In this case, we need to introduce a certain statistic and its corresponding threshold (hyper-parameter), which is hard to be determined in prior.
>
> (2) We have actually considered using the normalized entropy of the sampled samples as the statistical value for cut-off. As shown in Table below (Figure 5 in the revised version for details) , we found that this method not only has the hyper-parameter problem mentioned above, but also has no advantage over ESC in terms of performance-cost tradeoff (they both outperform SC). We believe this is because examining the cut-off point after each single sampling is too frequent, introducing greater randomness. This makes the model more likely to early-stop without sufficient sampling.
>
> Considering the above reasons, we chose the strategy presented. We have added this discussion in the revised version (A.4), thank you for your comments.

---

> > ### Comment · Reviewer_J879 · 2023-11-22
> >
> > Dear authors,
> >
> > thank you for taking the time and addressing the reviews, including the revision of your paper.
> >
> > I acknowledge the response and improvements in the paper towards my comments.

---

### Official Review · Reviewer_qSHs · 2023-11-06

**Soundness:** 3 good
**Presentation:** 3 good
**Contribution:** 2 fair
**Rating:** 5
**Confidence:** 5

**Summary:**

This paper presents Early-Stopping Self-Consistency (ESC), an adaptation of the original self-consistency to reduce the sampling cost. Instead of generating all samples at once, ESC generates samples in multiple sampling windows, and stops when all samples inside the same window produce the same results. They also provide a theoretical analysis on the sampling cost and the ESC performance compared to SC. They empirically evaluate ESC on multiple reasoning benchmarks, and demonstrate that ESC achieves comparable accuracies to SC, while the number of samples notably reduces.

**Strengths:**

1. ESC is a simple yet effective adaptation of the original self-consistency to reduce the sampling cost.

2. The ablation studies and theoretical analysis show that ESC is generally applicable to different benchmarks, and stays effective with different setups.

**Weaknesses:**

1. The novelty of this work is unclear. [1] already proposed an adaptation of self-consistency to reduce the sampling cost, but this work did not cite and discuss this prior work. Without a thorough discussion and direct comparison, it is unclear whether ESC is more effective.

2. In Table 1, when comparing ESC and L-SC, the performance difference is generally small. The reason can be that the improvement of SC saturates when the sample size increases, thus reducing the sampling size also does not drastically degrade the performance for SC. It is helpful to show this comparison for smaller sampling sizes, e.g., those in Table 2, and see if the performance improvement achieved by ESC can be more significant.

3. There are some issues in Table 1. For example, the SQA results of L-SC are generally much higher than SC, which look problematic. Also, it is confusing to list L in the table without additional notes, as L represents the sample size, while all other rows represent the task accuracies.

[1] Aggarwal et al., Let's Sample Step by Step: Adaptive-Consistency for Efficient Reasoning with LLMs, EMNLP 2023.

**Questions:**

1. Please clarify the novelty of this work. In particular, discuss and compare the approach to [1].

2. Show this comparison in Table 1 for smaller sampling sizes, e.g., those in Table 2, and see if the performance improvement achieved by ESC can be more significant.

3. Explain and fix issues in Table 1. For example, the SQA results of L-SC are generally much higher than SC, which look problematic. Also, it is confusing to list L in the table without additional notes, as L represents the sample size, while all other rows represent the task accuracies.

[1] Aggarwal et al., Let's Sample Step by Step: Adaptive-Consistency for Efficient Reasoning with LLMs, EMNLP 2023.

---

> ### Author Response · Authors · 2023-11-18
> **Response to reviewer qSHs (1/2)**
>
> Thanks for your detailed feedback! We will address your comments below:
>
> **Comparison with Adaptive-Consistency and the novelty of our work**
>
> Thanks for reminding and we will add citations to [1] and add more discussions on the novelty of our method:
>
> (1) Having additional control scheme
>
> Besides basic early-stopping self-consistency (ESC), we also propose its control scheme to dynamically choose the performance-cost balance. As we all know, along with the number of samples increases, performance will increase as well. For the realistic requirements, it is more practical to choose the affordable sample size rather than just try to maximize performance. We have derived the expectation of $\hat{L}$ (sampling number) and the upper bound of $\mathbb{E}(Q)$ (inconsistent probability). The experiments in Section 3.3 have shown that the predictions we obtain based on Equation 10 and Equation 14 are highly reliable for balancing sampling cost and voting performance.
>
> (2) Less sampling cost
>
> Adaptive-consistency (AC) generates samples step by step, which means each sample requires one input. Considering the demonstrations for in-context learning (usually 8 examples) have a lot of tokens, it will cost quite a portion of the budget. By contrast, ESC generates samples in multiple sampling windows, thus samples within one window can share the same input. Table 7 in updated paper shows that ESC can get higher accuracy with less sampling cost comparing with AC. A quick preview is below:
>
> | Method      |    \# prompt | \# completion  | Cost | Acc|
> | :--------: | :--------:| :--: | :--: | :--: |
> | SC  | 496.9 |  2909.0   | 0.0084 | 85.69|
> | AC     |   4930.3 |  813.2  | 0.0087|85.66|
> | ESC      |    1469.2 | 1220.8  | 0.0052|85.67|
>
> (3) No hyperparameters
>
> It is necessary for AC to tune the hyperparameter, specifically the confidence threshold. But for proposed ESC, the stopping criterion needs no hyperparameter due to the most conservative strategy to maintain the performance, i.e., all the answers within a window are same. Thus ESC can be conducted directly for different tasks and models, without any validation set. Besides, section 3.2 has proved the robustness regrading to window size w and max sampling size L.
>
> (4) More solid theoretical guarantees
>
> We provide more solid theoretical guarantees than AC lying in two parts: firstly, we rigorously derive the theoretical upper bound of the probability being inconsistent with the case where sampling size is infinite (true distribution of the model predictions according to Equation 3). While AC only considers the condition with next n samples, which is less rigorous; secondly, the upper bound of ESC is much more tight than AC. According to Equation 8, the bound value is only in the interval of $\leq 4 \times 10^{-2}$ to $\leq 2 \times 10^{-5}$ when window size varies from 3 to 10. By comparison，the confidence threshold of AC is chosen as 95\%.
>
> **Experiments with smaller sampling size**
>
> We thank you for your insightful suggestion and we conduct experiments with smaller sampling sizes (L). From the results in updated paper, Table 8 in Appendix A.3, the absolute value of performance drop is still not significant.
> We believe there are two reasons for this phenomenon: (1) In the paper of vanilla self-consistency [2], the large improvements come from the comparison between single greedy sample and 40 samples. The author plotted the accuracy with respect to varying numbers of samples. Within a certain interval, the performance change is not significant either. (2) While the performance drop of Table 1 remains insignificant, the corresponding reduction of L decreases a lot compared with table 1, which proves that the interval with smaller L is more unsaturated. A quick preview is below:
>
> |Method|CSQA L=10|CSQA L=20|GSM8K L=10|GSM8K L=20|
> |:-:|:-:|:-:|:--:|:-:|
> |SC|77.63|77.93|84.10|85.15|
> |ESC|77.63|77.91|84.10|85.15|
> |$\hat{L}$|6.47|8.60|7.09|10.14|
> |$\hat{L}$-SC|77.39|77.55|83.31|84.10|

---

> ### Author Response · Authors · 2023-11-18
> **Response to reviewer qSHs (2/2)**
>
> **Fixing issues in Table 1**
>
> We apologize that we have filled in the inconsistent values about SQA. That is because the candidate answer for question in SQA is either 'True' or 'False', but there will be some cases that the model generates neither of them. So there are two ways to vote the final answer: one is regarding these noisy answers as another type, e.g. '-1', to be included for voting; the other is removing this type of answers and only voting 'True' or 'False'. Our first choice was to vote based on the former criterion, but later we realized that the latter one made more sense, but we have forgotten to correct some values. We have fixed this problem in the revision.
>
> As for the representation of L in Table 1, thanks for your advice and we have modified Table 1 to make it more reasonable for reading.
>
> [1] Aggarwal et al., Let's Sample Step by Step: Adaptive-Consistency for Efficient Reasoning with LLMs, EMNLP 2023.
>
> [2] Wang et al., Self-consistency improves chain of thought reasoning in language
> models, ICLR 2023.

---

> > ### Comment · Reviewer_qSHs · 2023-11-23
> >
> > Thank the authors for the response and additional experiments.

---

### Author Response · Authors · 2023-11-21
**We are looking forward to your responses**

We appreciate the time and effort the reviewers and AC have put into reviewing our paper. We have thus submitted our responses to the reviewers' comments and questions. We hope to receive more feedback in the discussion to further improve the paper. Please take a moment to read our responses and let us know if you have any further comments/questions. Thanks.

---

### Author Response · Authors · 2023-11-23
**Summary of updates**

We appreciate the reviewers' effort in reviewing our paper. Their insightful comments can make our paper more solid and comprehensive. Below is a summary of new results and updates made according to the opinions from all reviewers. Please refer to each response for details.

**Add comparison with Adaptive-Consistency (AC) in Appendix A.2 and Related Work**

(per request from reviewer qSHs and UrCF)

**Modify description about Table 1 and add experiments with smaller sampling size in Appendix A.3**

(per request from reviewer qSHs and UrCF)

**Fix issues in Table 1**

(per request from reviewer qSHs and UrCF)

**Discuss on the choice of introducing observation window for the design of early-stopping strategy in Appendix A.4**

(per request from reviewer J879)

**Add comparison with state-of-the-art reasoning method in Appendix A.5 and show that ESC is actually an orthogonal method to other strong reasoning baselines in Appendix A.6**

(per request from reviewer rrBP)

We do hope that we have addressed all the comments. Since there is limited time for author-reviewer discussions, please discuss with us as soon as possible if there are any further questions or comments. Thanks again for valuable time and feedback!

---

### Meta-Review · Area_Chair_3KR6 · 2023-12-11

**Metareview:**

This work proposes a cost-effective strategy for self-consistency based prompting. The proposed method works pretty well in the sense that it can reduce the cost of naive self-consistency up to ~80% while pretty much retaining accuracy. Most reviewers and I have found the proposed method valuable, however, given the simplicity of approach and existence of some related ideas, the novelty of this work is not as strong. I believe this work compensates for this through its technical merits. For instance, the authors provide theoretical upper bounds of inconsistent answers (although I am not sure if the analysis/assumptions capture what happens in practice). They also provide failry thorough analysis including robustness aspect, contollable ES, and during rebuttal, integration with other prompting methods such as PHP. Overall, this is a borderline paper but I believe it can make a decent contribution to the ICLR venue.

**Justification For Why Not Higher Score:**

This paper has limited technical innovation to be considered as spotlight or oral.

**Justification For Why Not Lower Score:**

I am OK if the paper is rejected. It is really borderline.

---

### Decision · Program_Chairs · 2024-01-16

Accept (poster)